# Using Real Building Energy Use Data to Explain the Energy Performance Gap of Energy-Efficient Residential Buildings: A Case Study from the Hot Summer and Cold Winter Zone in China

**Xia Wang** [1], **Jiachen Yuan** [1], **Kairui You** [2,*], **Xianrui Ma** [3] and **Zhaoji Li** [4]

1     School of Public Finance and Taxation, Southwestern University of Finance and Economics, Chengdu 611130, China
2     Faculty of Architecture and the Built Environment, Delft University of Technology, 2628 BL Delft, The Netherlands
3     College of Economics and Management, Southwest University, Chongqing 400715, China
4     School of Management Science and Real Estate, Chongqing University, Chongqing 400044, China
*    Correspondence: K.You@tudelft.nl

**Abstract:** The International Energy Agency (IEA) emphasizes that using real building energy use data (RBEUD) to reflect the actual condition of buildings and inform policy-making is the most effective way to reduce buildings' carbon emissions. However, based on IEA's evaluation, regional and national building stock data are limited and lacking. Especially for China, the lack of RBEUD in buildings has limited our ability to address the energy performance gap (EPG). In this research, EPG refers to the difference between regulated energy consumption by design standards and actual energy usage. EPG makes it difficult to develop buildings that are energy-efficient. Therefore, this study aims to gather and analyze RBEUD in order to understand the role of occupants' behavior in explaining the EPG of energy-efficient residential buildings in China. The results suggest that the actual consumption of residential buildings is less than 1/5–1/3 of the theoretical limits. The heat pump and air conditioner's actual schedules and setpoint settings are the significant drivers that explain the EPG. In addition, the presentation of a database of 1128 households provides actual usage behavior parameters for policy-makers to improve the accuracy of building energy forecasting models.

**Keywords:** real building energy use data; energy performance gap; energy-efficient residential buildings; occupants' behavior

## 1. Introduction

The building represents the last mile sector in the global carbon neutrality transition [1]. It is considered to have the most substantial decarbonization potential using current strategies and technologies. Emissions from the building sector need to be reduced by approximately 40% between 2020 and 2050 to achieve global net-zero emissions by 2050 [2]. The International Energy Agency (IEA) predicts that more than 85% of global buildings need to meet zero-carbon-ready building energy codes by 2050 [2]. It also indicates that more and more energy-efficient buildings (e.g., green, low-carbon, sustainable, net-zero buildings) will emerge worldwide to reduce energy use and achieve global net-zero emissions. However, many studies have revealed a large gap between the designed and the actual energy consumption in energy-efficient buildings [3,4]. This gap is known as the 'energy performance gap' (EPG). The EPG would erode the credibility of the building and construction industry's policy-makers, designers, and engineers, and would also lead to general public skepticism of the new energy-efficient building concept [5]. The extensive literature on the EPG shows that attention should be paid to the role of occupant behavior

in shaping building energy consumption [6–8]. Therefore, collecting real building energy use data (RBEUD) of the building's operations to provide designers with valuable feedback on its actual performance is necessary to address EPG. IEA defines the dataset that could reflect the characteristics of buildings after they are in use as RBEUD [9]. The RBEUD includes energy use data, building features (such as floor area, year built, etc.), building morphology, household characteristics (such as the number of residents, household income, etc.), information on occupants' behavior (such as space heating, space cooling, window opening behaviors, etc.), and characteristics of household appliances (such as the energy-efficiency rating of appliances, etc.).

However, the lack of rich RBEUD in the building stock sector is the main barrier to reducing EPG worldwide [9]. Especially for China, the lack of RBEUD has limited our ability to address the EPG [1]. This makes it difficult to learn lessons to develop buildings that are more energy-efficient. The EPG has been highlighted in developed countries using RBEUD in recent years. For instance, in the UK, Menezes et al. [4] found that uncertain parameter assumption about occupancy behavior in building energy prediction models is the causal factor influencing EPG in buildings. Gill et al. [10] displayed that occupants opened windows more frequently than model assumptions. Thus, the low-energy design must address issues with factors of occupants' behavior more adequately. Gupta et al. [11] found that occupant factors associated with higher demand temperatures, frequent window openings in winter, and the over-use of heating systems were responsible for the EPG between modeled and actual energy use. In Demark, Carpino et al. [12] suggested that standard occupancy schedules indicate high heat gains, which lead to an underestimation of energy use for space heating or cooling. The cause for the EPG in Ireland is higher actual indoor temperatures than those assumed in the model [13], Dutch dwellings [14], and Swiss dwellings [15]. In Canada, Rouleau et al. [16] suggested that the two key variables that justify the EPG are the set point temperature and the control of windows in energy-efficient social housing buildings. In Germany, Galvin [17,18] emphasized that incorrect assumptions (both standardized occupancy and technical factors) could be linked to the EPG. In China, Liang et al. [19] suggested that a lack of sub-metering for air conditioners and longer operating hours cause the EPG in green commercial buildings. Wu et al. [20] provided design factors referring to values from local standard authorities that led to the overdesign of green office buildings. However, few studies have focused on issues regarding the EPG of energy-efficient residential building sectors in China. In addition, updated information about the energy-efficient residential buildings in use is still necessary for improving the accuracy of building energy consumption models which can support policy making. Therefore, to fill the gap in the literature, the goals of this study include: ① gathering and analyzing RBEUD at the household level to identify and quantify the EPG in the residential sector; ② quantifying the reasons behind the EPG by comparing the occupant behaviors between design consumption and operation conditions. This research not only focuses on quantifying the gap, but also identifies the factors influencing the difference in energy consumption at high, medium, and low consumption levels. This can improve our understanding of the energy-efficient residential buildings currently in use in order to help reduce the EPG in residential buildings in the future.

This research makes several contributions to the literature. ① It enriches the residential building stock database in China. Rich datasets, including actual energy usage and household behavior data, are used to measure the EPG at the household scale, avoiding single buildings with limited heterogeneity. ② It provides a significant empirical contribution to the literature by using occupants' behavior pattern data to explain the EPG in the residential building sector in China. ③ Based on empirical evidence, this study demonstrates that underconsumption (actual consumption less than theoretical) situations could explain the prebound effect (actual savings less than theoretical) in residential buildings. The above results contribute to a better understanding of the current state of residential energy consumption, provide input parameters for building codes and standards, and offer some policy suggestions for building energy efficiency.

This research is structured as follows. Section 2 summarizes the literature review. Section 3 presents the method. Section 4 evaluates the building energy performance. Section 5 discusses and quantifies the EPG. Section 6 concludes the paper and offers some policy suggestions.

## 2. Literature Review

According to the literature on the EPG, the difference between predicted and actual building energy consumption can be explained by three categories: design-related, construction-related, and operational-related factors. It is noted that unrealistic predictions in the design stage are one of the most critical factors causing the EPG [4,5,21–23]. Predicted energy consumption is based on the input data of the building energy model. For example, the assumed building operation indicators mainly include the air conditioning/space heating schedule, ventilation rates, setpoint settings, appliance energy efficiency levels, etc. Thus, with limited RBEUD, it is difficult for a designer to accurately predict the operation consumption [21,24,25]. Currently, data regarding the actual energy performance of existing building stock worldwide are significantly limited and lacking [1]. According to IEA, if sufficient data can be collected during the operation phase, the design will be more precise. Otherwise, a persistent lack of such data will likely result in an increasing gap between theory and practice, which will lead to a failure to achieve the net-zero emissions goals.

In the construction stage, the quality of the onsite construction may cause the EPG. Some common examples which may lead to a gap between the design and the actual building include: ① insulation gaps and thermal bridging are rarely considered in the building energy simulations [21]; ② a contractor with limited experience and knowledge may not fully understand the designer, which may result in a gap between the design and the as-built result [26]; ③ in China, occupants renovate 51% of residential buildings. However, approximately 87% of the occupants do not have enough knowledge about building envelope performance, which may destroy external thermal insulation systems [27].

In the operation stage, occupants' behaviors significantly impact buildings' actual energy consumption [28]. As a result, assumptions about occupant patterns and behavior, which are inherently unpredictable and highly uncertain, inevitably lead to significant uncertainty in energy forecasting [29]. In reality, over-consumption or under-consumption phenomena exist due to occupants' behaviors. For example, Cozza et al. [30] and Guerra-Santin and Itard [31] confirmed the phenomenon of overconsumption in energy-efficient buildings. They found that the energy demand for residential buildings was mostly influenced by occupants' behaviors, and that occupant behaviors could explain 3.2–9.4% of the change in energy consumption. Majcen [32] found results regarding behavior that were able to explain 9.1% of the variance in energy consumption. On the other hand, Galvin and Sunikka-Blank [33] demonstrated that restriction behaviors could lead to under-consumption phenomena in energy-efficient buildings. The actual consumption was calculated as 30% below the calculated consumption based on German datasets [33,34]. Loga et al. [35] found that the actual consumption was 31% lower than the calculated energy performance. Similar results were also obtained in the latest studies [30,36].

In summary, many studies have analyzed reasons for the EPG from various aspects. However, limited empirical evidence explains the EPG. Thus, this research aims to contribute to the literature on the EPG by using occupants' behavior pattern data to explain it in the residential building sector in China.

## 3. Methodology

### 3.1. Overall Research Approach

To understand and capture the actual household energy consumption and energy usage behavior of energy-efficient residential buildings in a holistic way, the general research approach is given in Figure 1. ① A questionnaire survey was conducted in Chongqing, China. Residential buildings are complex systems, and various factors influence actual building energy performance. For instance, building-related characteristics, residents' oc-

cupancy information, and real energy use data were collected through a survey. ② This research uses three methods for data analysis: the Lorenz curve and Gini coefficient, the chi-square test, and gap analysis. The Lorenz curve and Gini coefficient was applied to capture characteristics of electricity use across households. The chi-square test was used to test the statistical differences in the distribution of the lowest, the average, and the highest energy-consuming groups for the household's occupant behavior characteristics. The gap analysis was used to analyze the root cause of the EPG in residential buildings.

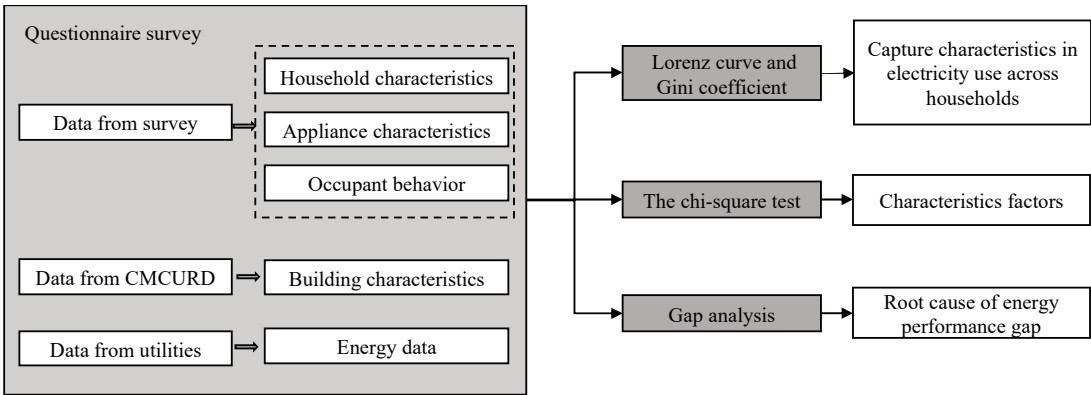

**Figure 1.** Analysis approach for the survey results. Notes: CMCURD: Chongqing Municipal Commission of Urban-Rural Development.

### 3.2. Survey Description

The household survey adopted the model of cooperation between the government (Chongqing Municipal Commission of Urban-Rural Development, CMCURD), utilities (the State Grid Chongqing Electric Power Company: Chongqing, China, SGCEPC), and colleges and universities (Chongqing University). The data collection process consisted of three stages (see Figure 1). ① The investigators of Chongqing University conducted a door-to-door survey in the sampled region. They explained the purpose and the content of the questionnaires to the respondents, accompanied by the employees of the property management company. The respondents filled in the questionnaires and returned them on-site. Families participating in the survey needed to meet the following criteria. First: the household could provide electricity consumption billing or the smart meter number. Second, the household's electricity consumption was mainly for living rather than the business. Third, the family must have lived in the residence for more than 12 months before 2016. ② We received information about the buildings' characteristics (including the level of building energy efficiency standards (BEES) used for building construction and the year of construction) from CMCURD. ③ We collected data from SGCEPC using the smart meter numbers or electricity billing numbers.

This household-level database covered five modules: building characteristics, households, appliances, occupant behavior characteristics, and electricity data. Building samples were distributed in all nine districts of Chongqing city (See Figure 2). Eventually, after removing outliers from the database, this dataset contained 43 communities and 1128 households in the Chongqing urban area. Of those communities, 70% were ordinary housing estates, 14% were upscale residential districts; 7% were eco-districts, and 9% were indemnification housing (see Figure A1). Collecting samples from these different property forms of the communities can represent different income groups. The total floor space of these 43 communities covers 10.88 million m$^2$, and this guaranteed the representativeness of the respondent sample.

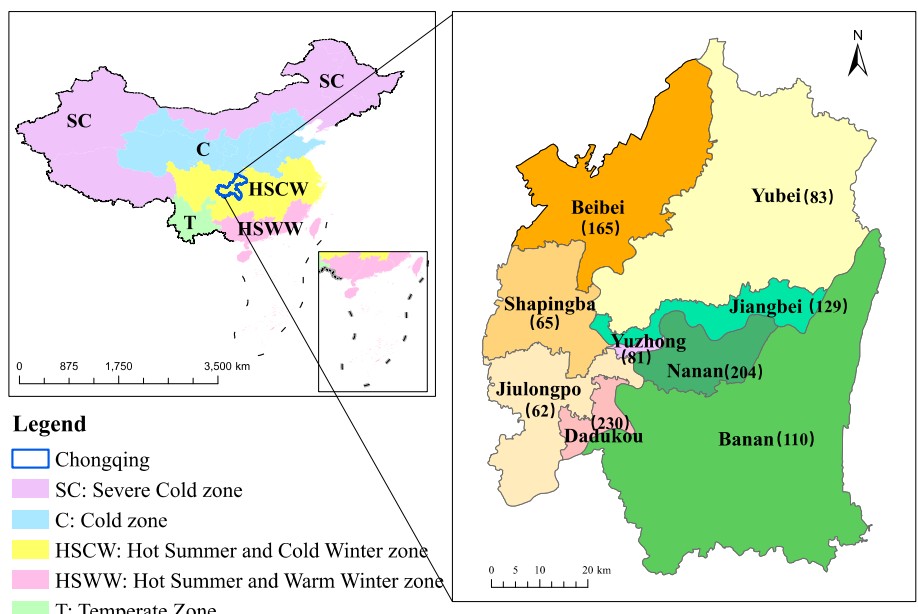

**Figure 2.** The survey area and its location. Note: The numbers in parentheses represent the size of samples collected in each district.

To represent the level of building performance, the database divided the buildings into three categories: ① Buildings were built based on the 'Design Standard for energy efficiency 50% of residential buildings JGJ134-2001' (hereafter, 50%-BEES). ② Buildings were built based on the 'Design Standard for energy efficiency 65% of residential buildings JGJ134-2010' (hereafter, 65%-BEES). ③ Buildings were built without energy-efficiency measures (hereafter, IBs). We defined residential buildings built with 65%-BEES and 50%-BEES as energy-efficient residential buildings. All of the energy-efficient residential buildings were constructed between 2002 and 2014, while the IBs were built before 2002. This database contained 508 residential buildings built with 65%-BEES and 338 residential buildings built with 50%-BEES. The number of IBs was 282. The average floor area of all investigated households in Chongqing was 92 m per household (m$^2$/household). In addition, 67% of households were situated in residential buildings with floor areas between 60 and 120 m$^2$ (see Figure A2). The floor area per capita was 30.23 m$^2$.

The average household size of the surveyed residents was 3.44 people, which was higher than the municipal average household size of 3.13 people as well as the nation's statistical household size of 3.11 people in 2016 (See Figure A3). This is because our survey excluded non-family households, such as army barracks and college dorms. In total, 76% of the households consisted of three and more people, as shown in Figure A4. With the second-child policy, the family structure showed a preference to extend families to consist of two adults living with their children and parents.

The database divided people into five income classes. The annual household income structure in the survey is presented in Figure A5. Of the respondents, 14% were from low-income households (less than CNY 30,000). The lower-middle income (CNY 30,000–60,000) and middle-income (CNY 60,000–120,000) households accounted for 29% and 38% of the total surveyed households, respectively. Of those 15%, had upper-middle income levels (CNY 60,000–120,000), and 4% were high-income households (more than CNY 200,000). Different income levels represented different levels of energy consumption, to some extent.

The ownership of major household appliances in our dataset compared to the national and municipal information is shown in Figure A6. The ownership rate of split heat pump air conditioners (ACs) in our survey data was 302 sets per 100 households, higher than the national and municipal levels. There are two main reasons for this: first, Chongqing has hot summers and a humid climate, and split heat pump ACs are the main equipment for

cooling and heating. Second, the survey area is the most developed region of Chongqing. The residents are relatively wealthy and have high requirements for indoor comfort.

## 4. Measurement of the Actual Buildings' Energy Performance

In this section, residential buildings' energy consumption is analyzed based on the actual consumption data, with a focus on space heating and cooling.

### 4.1. Actual Building Energy Consumption

Figure 3 shows the distribution of the survey samples' annual household electricity consumption in 2016. The annual household electricity use varied from 238 kWh/household to 12,942 kWh/household. The average household use was 2573 kWh/household. There was a significant difference between the lowest and the highest energy consumption. Therefore, we needed to conduct further studies on the highest and lowest groups. This study employed the Lorenz curve and Gini coefficient to demonstrate the characteristics of electricity consumption across households. The traditional Lorenz curve and Gini coefficient are the most commonly used methods in economics for analyzing income inequality. Following [37], we used them to analyze inequality in energy consumption. The energy Gini coefficient is defined as:

$$Gini = 1 - \left| \sum_{i=1}^{N} (X_{i+1} - X_i)(Y_{i+1} - Y_i) \right|$$

where $X$ refers to the cumulative proportion of the household, calculated by the number of households in population group $i$ divided by total households. $Y$ refers to the cumulative proportion of electricity consumption. $Y_i$ equals the electricity consumption of the $i_{th}$ household divided by the total electricity consumption, which is ordered from lowest to highest.

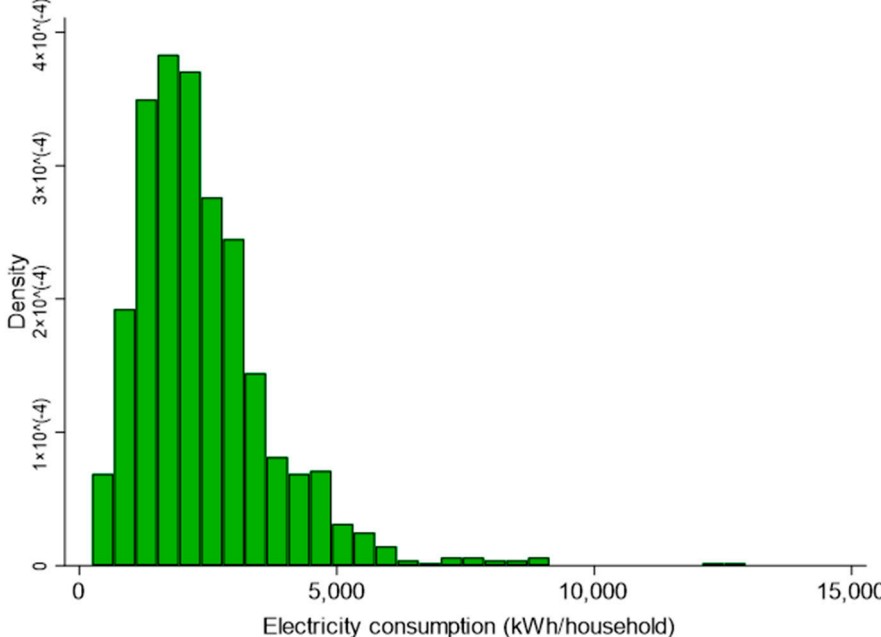

**Figure 3.** Distribution of electricity consumption per household. Notes: The horizontal axis refers to the annual household electricity consumption in 2016, expressed in kWh. The vertical axis refers to the density.

Figure 4 shows the Lorenz curve of electricity consumption across households. The Gini coefficient of 0.284 means that the top 10% of surveyed urban residential households were responsible for more than 20% of electricity consumption. According to the Lorenz

curve, household electricity consumption may be divided into three groups: the lowest 10%, the average 80%, and the highest 10% electricity-consuming groups. A more detailed comparison was made to better understand the reasons by comparing the occupants' behavioral characteristics in the three groups. The chi-square was used to test for statistical differences between the three groups.

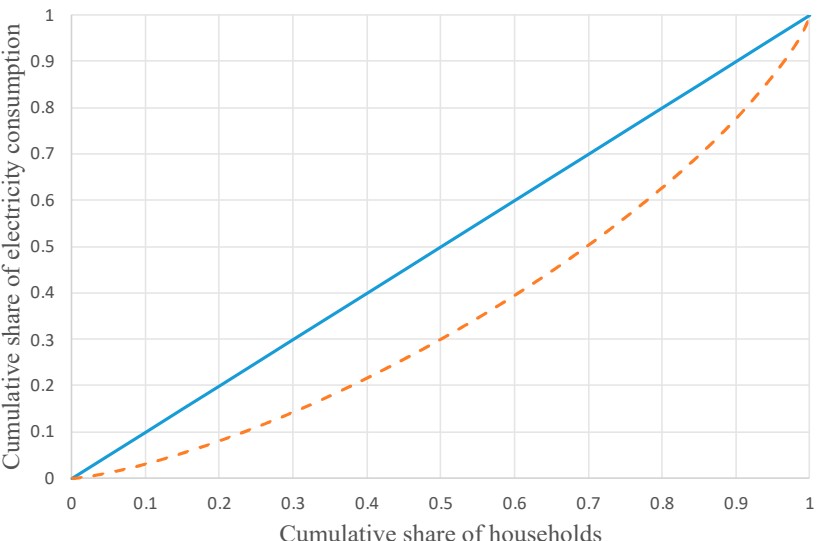

**Figure 4.** Lorenz curve of electricity. Notes: The blue diagonal is the line of perfect equality. The orange dotted line indicates electricity consumption in 2016.

The annual household income levels of the highest 10%, the average 80%, and the lowest 10% electricity-consuming groups were compared. The distribution of annual income was different among each group (Pearson chi2(8) = 43, $p = 0.000$) (see Figure 5). This finding was consistent with the research results by Zhou et al. [38] and Chen et al. [39], who showed that income had a positive effect on building energy use using household survey data. Lower-income (less than CNY 30,000) families were found more frequently in the lowest energy consumption group. At the same time, higher-income households were more likely to belong to the highest energy-consuming group. This finding implies that urban residential electricity demand increases with income growth.

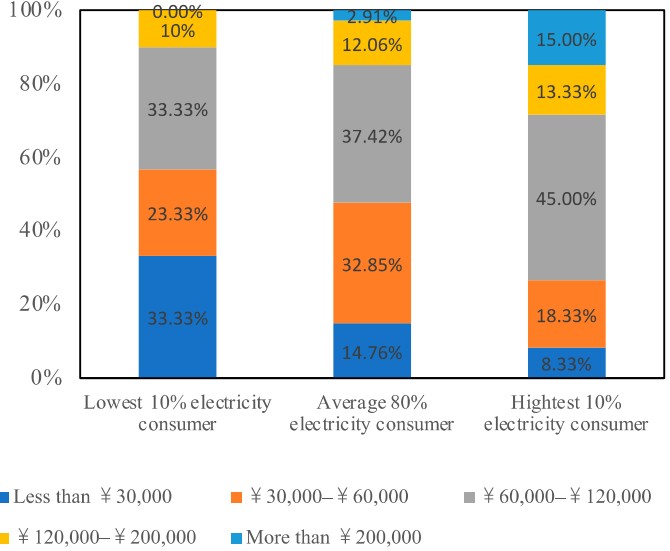

**Figure 5.** The distribution of annual income for the lowest 10%, average 80%, and highest 10% electricity-consuming groups.

The chi-square test revealed a significant variation in the distribution of the four floor space categories among the three electricity-consuming groups: highest, lowest, and average (Pearson chi2(6) = 62, *p* = 0.000) (see Figure 6). Large floor spaces tended to be found in the highest electricity-consuming group. This finding is consistent with the previous studies [40,41].

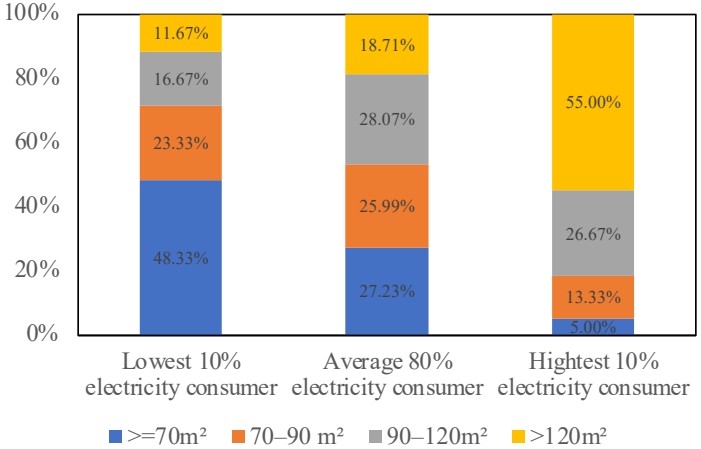

**Figure 6.** The distribution of floor space for the lowest 10%, average 80%, and highest 10% electricity-consuming groups.

Many previous studies have identified that the number of household members is an essential factor influencing electricity use [29,42]. In agreement with them, extended families were found more frequently in the high electricity-consuming group. The distribution differences between groups were significant (Pearson chi2(8) = 50, *p* = 0.000) (see Figure 7).

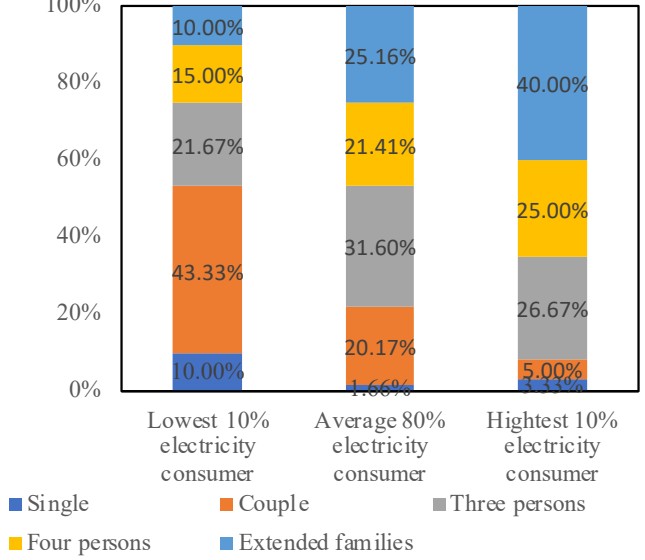

**Figure 7.** The distribution of occupants for the lowest 10%, average 80%, and highest 10% electricity-consuming groups.

Another discrepancy in electricity use between the lowest 10%, average 80%, and highest 10% electricity-consuming groups was the method of space heating and cooling (Figures 8 and 9). Obviously, central ACs contribute to high electricity use in the summertime. Households using under-floor heating in the highest 10% of electricity consumers were higher than those in the lowest 10% electricity-consuming group.

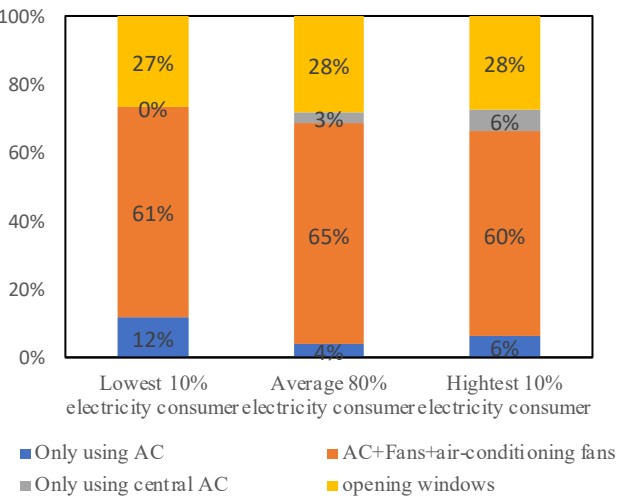

**Figure 8.** The distribution of cooling methods for the lowest 10%, average 80%, and highest 10% electricity-consuming groups.

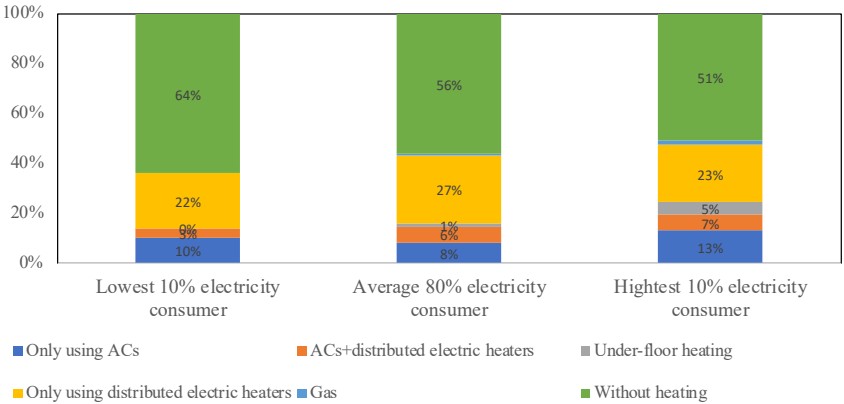

**Figure 9.** The distribution of heating methods for the lowest 10%, average 80%, and highest 10% electricity-consuming groups.

In conclusion, residential building electricity use varies between households and with occupant behavior characteristics. The common features of the lowest electricity-consuming households were lower income, smaller floor area, single or couple residents, and a lack of central air conditioning and underfloor heating. The common characteristics of the highest electricity-consuming group were higher income, larger floor area, and extended families living in the household. They also showed full-time and full-place usage patterns of heat pump ACs. The high electricity-consuming group is a good illustration of a full-time and full-place lifestyle. However, in China, most people's lifestyles currently consist of part-time and part-place usage of heat pump ACs [43].

Figure 10 demonstrates annual household electricity usage by different levels of BEES. In 2016, annual electricity consumption in 65%-BEES households varied from 238 kWh/household to 9125 kWh/household, and the average value was 2249 kWh/household. Yearly electricity consumption in 50%-BEES households varied from 245 kWh/household to 8060 kWh/household, and the average value was 2132 kWh/household. For IBs, the annual household electricity usage ranged from 352 kWh/household to 12,942 kWh/household, and the average value was 3018 kWh/household.

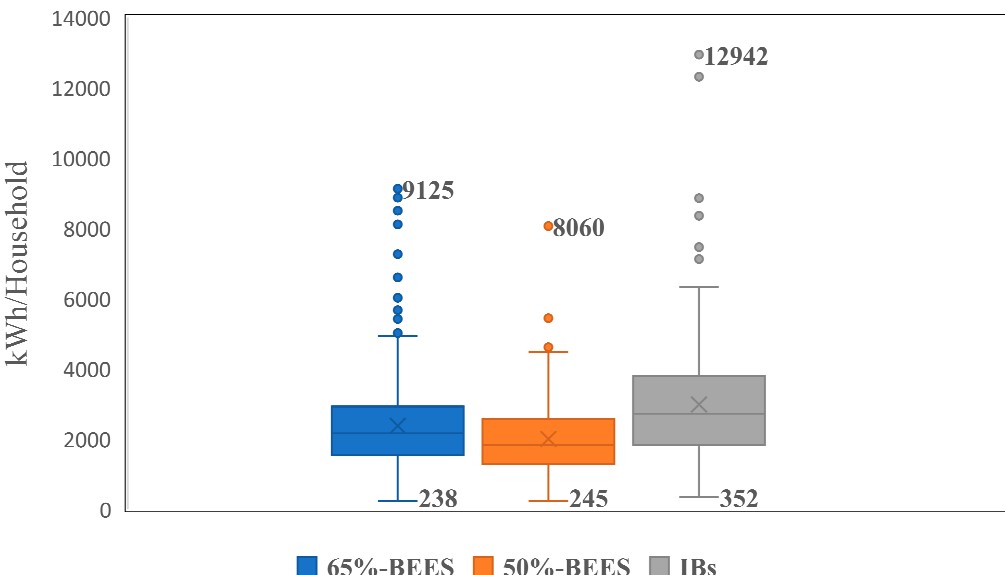

**Figure 10.** Total energy consumption in the three types of buildings.

Figure 11 shows total annual household electricity usage intensity (EUI, kWh/m$^2$/year) by different levels of building energy efficiency standards. In 2016, the EUI in 65%-BEES varied from 3.72 kWh/m$^2$/year to 104.33 kWh/m$^2$/year, and the average value was 26.01 kWh/m$^2$/year. The EUI in 50%-BEES varied from 3.82 kWh/m$^2$/year to 115.14 kWh/m$^2$/year, and the average value was 28.53 kWh/m$^2$/year. In IBs, the EUI ranged from 4.77 kWh/m$^2$/year to 102.78 kWh/m$^2$/year, and the average value was 28.08 kWh/m$^2$/year.

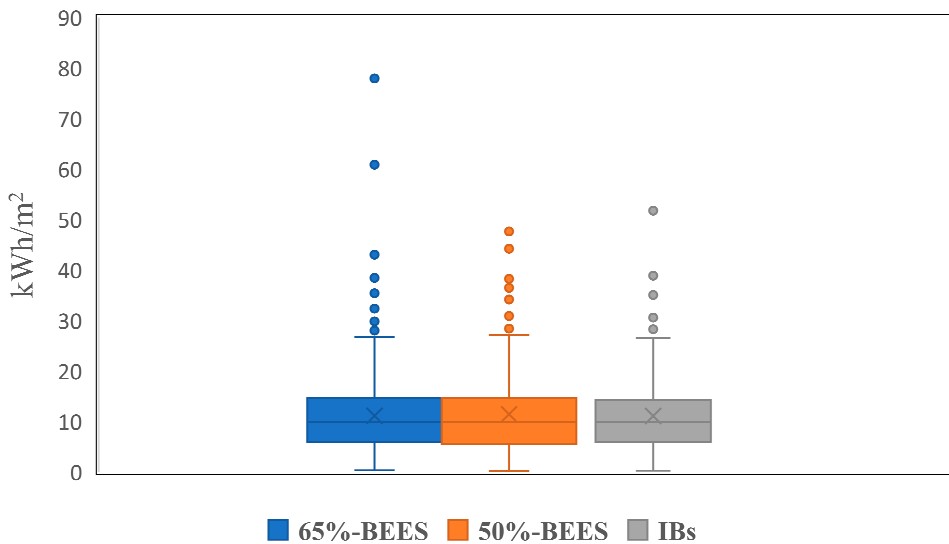

**Figure 11.** Energy intensity in the three types of buildings.

### 4.2. Actual Building Heating and Cooling Consumption

This section describes the statistics regarding space heating and cooling electricity consumption in energy-efficient buildings and IBs (See Table 1). The actual consumption of space heating and cooling was estimated according to the monthly electricity consumption obtained by the smart meter. We used an approximating method, as was introduced in the previous research [27].

**Table 1.** The statistics of space heating and cooling energy consumption in the three types of buildings.

| | IBs | | | 50%-BEES | | | 65%-BEES | | |
|---|---|---|---|---|---|---|---|---|---|
| | Mean (Std. Dev.) | Max | Min | Mean (Std. Dev.) | Max | Min | Mean (Std. Dev.) | Max | Min |
| Electricity consumption for space cooling (kWh/household) | 891 595.24 | 3833 | 18 | 559 464.76 | 2986 | 4 | 602 458.17 | 3399 | 4 |
| Electricity intensity for space cooling (kWh/m$^2$) | 8.28 5.67 | 33.44 | 1.08 | 7.33 5.74 | 30.18 | 1.06 | 6.72 4.79 | 29.8 | 1.05 |
| Electricity consumption for space heating (kWh/household) | 232 379.05 | 3813 | 0 | 151 256.63 | 2002 | 0 | 164 289.23 | 3463 | 0 |
| Electricity intensity for space heating (kWh/m$^2$) | 1.95 2.63 | 38.48 | 0 | 1.91 2.99 | 20.43 | 0 | 1.93 3.41 | 15.42 | 0 |
| The percentage of space cooling accounting for total buildings' electricity consumption (%) | 30% | | | 26% | | | 26% | | |
| The percentage of space heating accounting for total buildings' electricity consumption (%) | 8% | | | 7% | | | 7% | | |
| The percentage of space cooling and heating accounting for total buildings' electricity consumption (%) | 37% | | | 33% | | | 33% | | |

### 4.2.1. Electricity Consumption for Space Cooling

The average annual household electricity consumption for space cooling was 662 kWh/household, accounting for 27% of the total annual household electricity consumption. The average annual household electricity consumption for space cooling for IBs was 891 kWh/household, with a standard deviation of 595.24 kWh/household. The average annual household space cooling electricity consumption for 50%-BEES was 559 kWh/household, with a standard deviation of 464.76 kWh/household, while 65%-BEES buildings consumed 602 kWh/household, with a standard deviation of 458.17 kWh/household. The average cooling electricity intensity for IBs was 8.28 kWh/m$^2$/year, with a standard deviation of 5.67 kWh/m$^2$/year. The average cooling electricity intensities for 50%-BEES and 65%-BEES buildings were 7.33 kWh/m$^2$/a (with a standard deviation of 5.74 kWh/m$^2$/a) and 6.72 kWh/m$^2$/a (with a standard deviation of 4.79 kWh/m$^2$/year), respectively. The share of cooling electricity usage out of the total electricity usage for 65%-BEES and 50%-BEES buildings was 26%, and 30% for IBs.

### 4.2.2. Electricity Consumption for Space Heating

The annual average household heating consumption was 177 kWh, accounting for 7% of the total annual household electricity. The average annual household space heating electricity consumption for IBs was 232 kWh/household, with a standard deviation of 379.05 kWh/household. The average annual household space heating electricity consumption rates for 50%-BEES and 65%-BEES buildings were 151 kWh/household (with a standard deviation of 256.63 kWh/household) and 164 kWh/household (with a standard deviation of 289.23 kWh/household), respectively. The heating consumption varied greatly among households due to different living styles and heating habits. Households using central heating had a power consumption rate of 38.48 kWh/m$^2$, which is about 20 times the city's average.

## 5. Discussion

### 5.1. Energy Performance Gap Analysis

Figures 12 and 13 provide overall pictures of the average annual household electricity use and average electricity use intensity (EUI) of energy-efficient buildings and IBs in 2016. In Figure 12, the horizontal line represents the constraint value of energy use demonstrated in the 'Standard for Energy Consumption of Buildings (GB/T51161-2016)', which is the first standard for buildings' energy consumption in China. Two horizontal lines in Figure 13

represent two design values of EUI: (1) the EUI defined by the 'Design standard for energy efficiency 50% of a residential building of Chongqing, DB50/5024-2002' (50%-BEES); (2) the EUI defined by the 'Design standard for energy efficiency 65% of a residential building of Chongqing, DBJ50-071-2007' (65%-BEES).

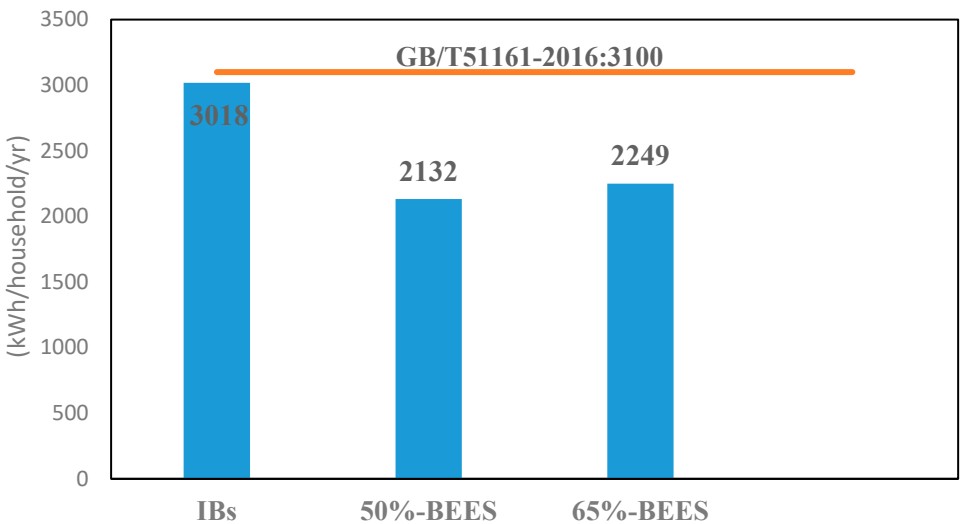

**Figure 12.** Average annual household electricity consumption in Chongqing.

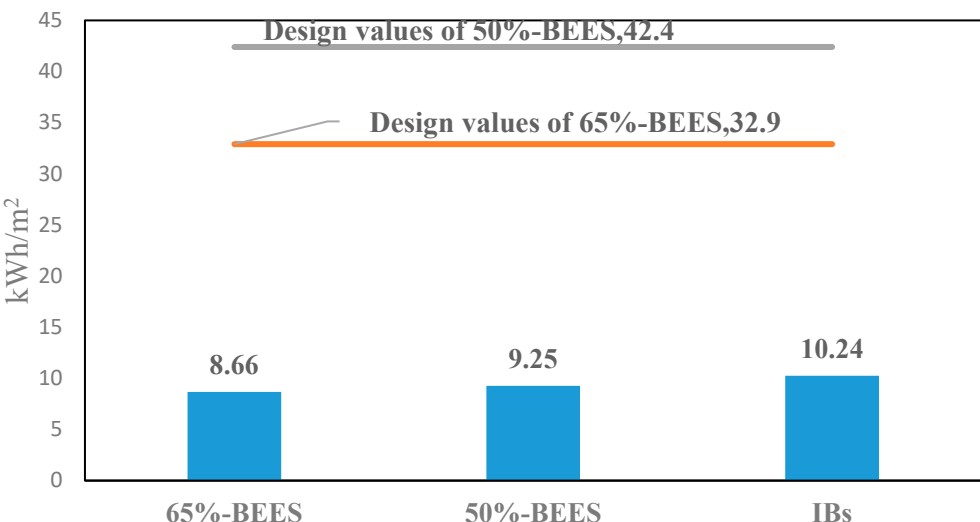

**Figure 13.** Actual cooling and heating electricity consumption and regulation limit values.

The average household electricity use of IBs was 3018 kWh/household in 2016. The average household electricity use of 50%-BEES was 2132 kWh/household in 2016. Finally, the average household electricity use of 65%-BEES was 2249 kWh/household in 2016. All were lower than the GB/T51161-2016 energy target of 3100 kWh/household. This means that most of the buildings met the constraint values.

The actual average heating and cooling EUI of energy-efficient buildings and IBs were much lower than the regulation limit values defined in DB50/5024-2002 and DBJ50-071-2007 (see Figure 13). The regulation limit value of 65%-BEES was 32.9 kWh/m$^2$, while the actual average heating and cooling EUI in buildings with 65%-BEES was 8.65 kWh/m$^2$, less than 1/3 of the regulation limit value. The regulation limit value of 50%-BEES was 42.4 kWh/m$^2$, while the actual average heating and cooling EUI in buildings with 50%-BEES was 9.25 kWh/m$^2$, less than 1/5 of the regulation limit value. It can be seen that the benchmark value of energy-efficient buildings is far greater than its actual electricity consumption value. This phenomenon is identified as the 'prebound effect' [34]. The pre-

bound effect is the opposite phenomenon of the rebound effect. The rebound effect tends to occur in low-energy dwellings, where occupants consume more than the theoretical amount of energy. Contributions from 'prebound' and rebound effects are likely to swallow up a significant portion of the calculated energy saving (see Figure 14). The prebound effect phenomenon has been recognized in recent Dutch [44], Belgian [45], French [46], and UK studies [40]. In China, little research has paid attention to the prebound effect of building energy efficiency policies [47]. The prebound effect will increase with the increase in the regulation limit values assumed by the building energy-efficiency standard [34]. This implies that using the design energy rating of energy-efficient buildings to predict energy savings tends to overestimate savings. The potential energy savings through non-technical measures such as occupant behavior may well be far larger than is generally assumed in policies; thus, policy-makers need a better understanding of what drives or inhibits occupants' decisions.

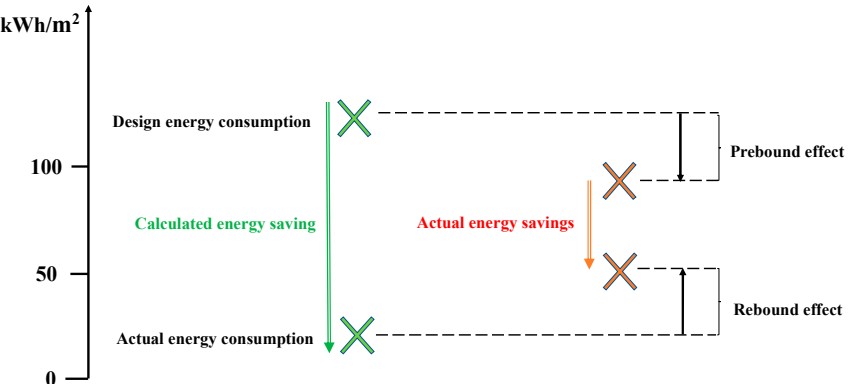

**Figure 14.** An illustration of how the prebound and rebound effects may limit energy saving, reducing it from its theoretical amount [34].

### 5.2. Reason behind EPG

The statistical results above suggest that a large EPG exists between the actual and designed total electricity consumption of energy-efficient buildings in Chongqing. The EPG is a difficult puzzle for policy-makers. Researchers and practitioners have attempted to propose strategies to reduce the EPG [18,29,48,49]. Based on the RBEUD for energy-efficient buildings, this section discusses the root cause of the EPG with varying occupant behaviors between design consumption and operation conditions.

Table 2 summarizes the difference in parameters between the theoretical lifestyle assumed in the BEES for energy-efficient buildings and the schedules of most residents from the survey. ① The assumed lifestyle in the standard can be described as 'Full-time and Full-place' use and application of electricity [50]. 'Full-time' represents heating and cooling systems operating all the time during cooling (June 1–September 30) and heating (from December 1 to February 28) seasons [51]. However, compared to this design condition, the actual schedules of heat pumps and air conditioners are very different according to the survey data. In Figure 15, 90% of the surveyed residents used their heating for a duration of fewer than 2 months in the wintertime, among which 47% of the families used their heating for a duration of fewer than 15 days, and only 9% of the families used their heating for a duration of 2–3 months. In terms of the average daily heating hours (See Figure 16), 83% of the households used heating time for fewer than 3 h per day. Turning to space cooling, 67% of the surveyed households used their cooling for fewer than 3 months, and 33% used their cooling for 4 months or more (See Figure 17). Of the surveyed households, 61% used ACs for cooling for more than 9 h a day, with a maximum time of 12 h (see Figure 18). 'Full-place' signifies cooling and heating systems operating in all areas of residential buildings. Nearly 93% of the surveyed households indicated that when they use ACs, they either limit the use to one bedroom or use ACs when somebody is in the room in the summertime. In the wintertime, 98% of families indicated that when they use

ACs, they limit the use to one bedroom. Above all, it can be inferred that at least 90% of the residents of Chongqing have adopted the 'part-time and part-space' lifestyle, and the proportion of residents who live according to the theoretical lifestyle assumed in the BEES accounts for less than 10% of the surveyed residents in this region. ② The ventilation rate defined in the BEES is 1 time/hour, whereas the ventilation rate is random in reality. People who live in Chongqing prefer to open windows. This behavior has resulted in a significant discrepancy between the predicted and the actual thermal insulation performance. As suggested in Figure 19, 32% of occupants open either windows or doors when they use AC. ③ The regulated AC temperature setpoint is 26 °C in the summertime and 18 °C in the wintertime according to the BEES. The actual temperature setpoint in summer is shown in Figure 20. Only 41% of households turn on the AC at 26 °C in summer. Less than 10% of households turn on the AC at 18 °C in winter. ④ The design condition energy efficiency labels (EELs) of heat pump ACs and the actual situation are different. According to Table 2, the standards show that the design condition assumption of energy efficiency ratio of ACs ranges from 2.2 to 2.8, while the EEL of the heating system ranges from 1.0 to 2.8 [51–54]. However, the actual EEL of heat pump AC equipment can be improved quickly; more than half of the surveyed households have heat pump ACs with EELs of grade 1.

**Table 2.** The design parameter regulations in BEES and the real conditions surveyed.

| | | **Building Operation Schedule (Lifestyle)** | | | | | **ACs and Heating System Features** |
|---|---|---|---|---|---|---|---|
| | | **Conditioned Space** | **Conditioned Period** | **Ventilation Rate** | **Temperature Setting** Summer | Winter | **EEL ACs/Heating** |
| Design Condition | IBs (Baseline) | All space | ACs: From 1 June to 30 September Heating: From 1 December to 28 Febuary the next year | 1 time/h | 26 °C | 18 °C | 2.2/1.0 |
| | 50%-BEES | | | | | | 2.3/1.9 |
| | 65%-BEES | | | | | | 2.8/2.8 |
| Actual Condition | IBs | Part-space | Part-time | Arbitrary | Diversified | Diversified | At least Grades 3 |
| | 50%-BEES | | | | | | |
| | 65%-BEES | | | | | | |

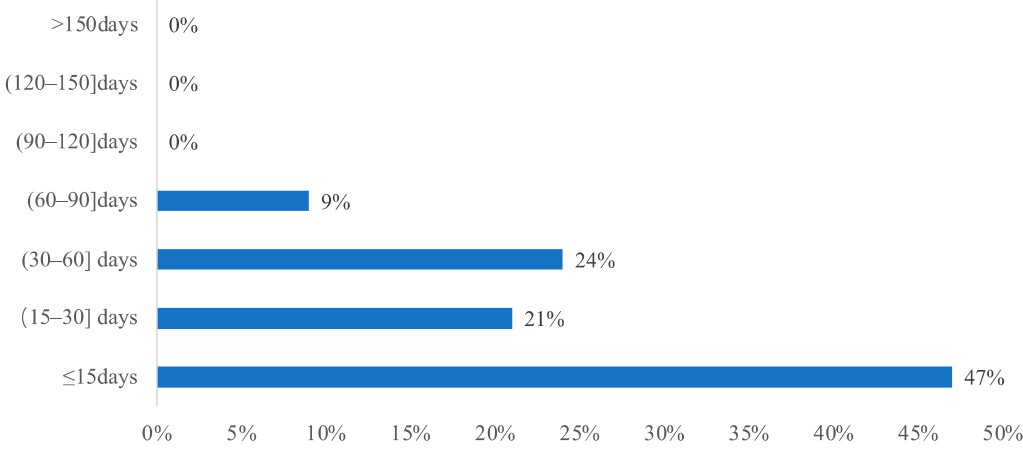

**Figure 15.** Average annual duration of heating usage in winter.

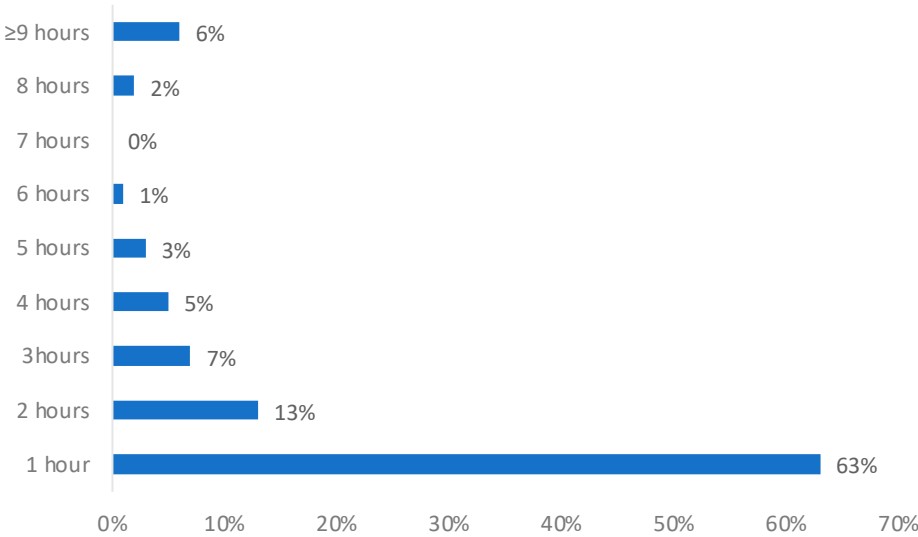

**Figure 16.** Average daily duration of heating usage in winter.

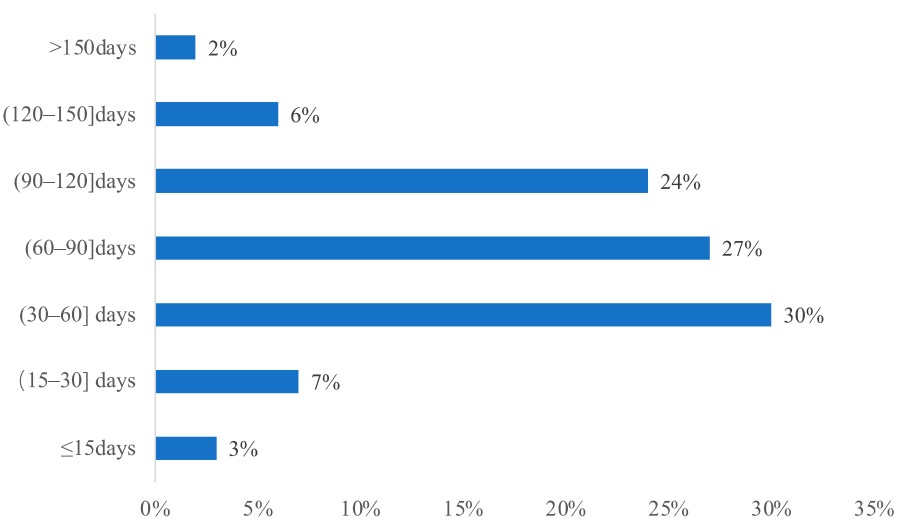

**Figure 17.** Average annual usage of ACs in summer.

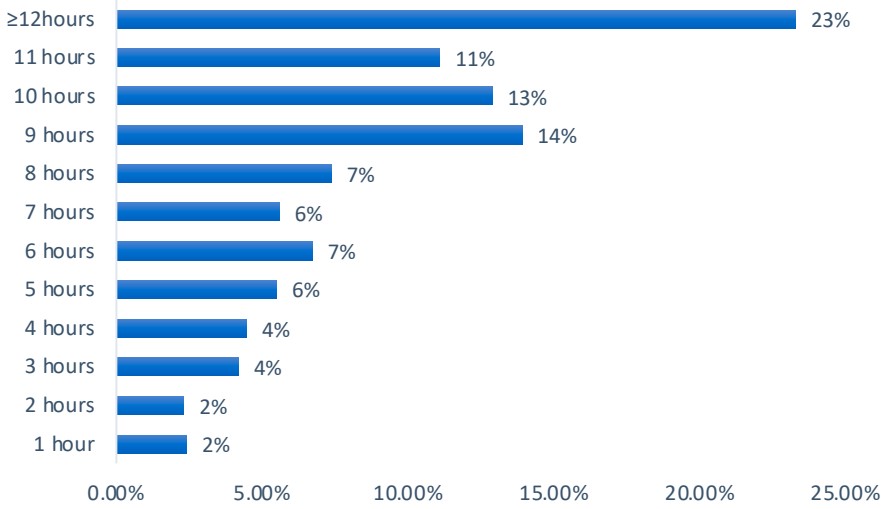

**Figure 18.** Average daily usage of split ACs in summer.

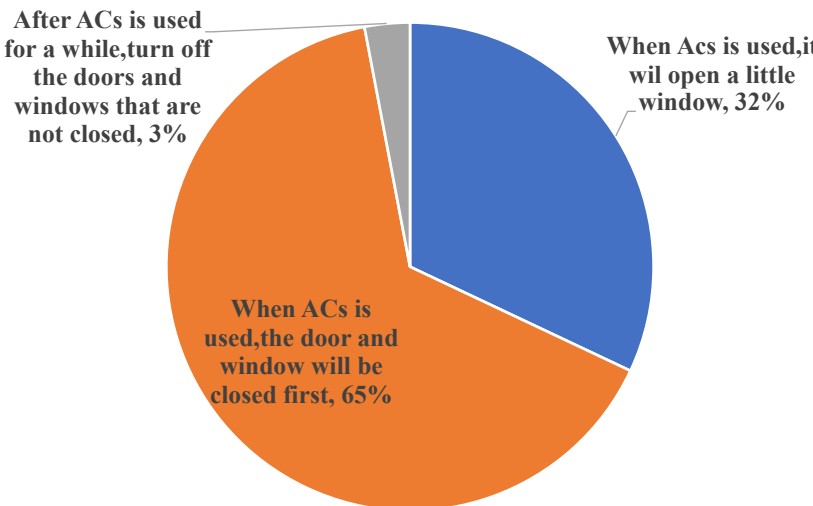

**Figure 19.** Behaviors regarding usage of windows.

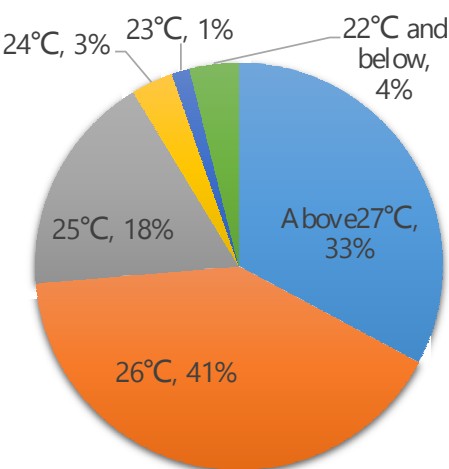

**Figure 20.** Air conditioner cooling setpoint temperatures in summer.

## 6. Conclusions

This study provides empirical evidence to quantify the EPG between theoretical and actual consumption of energy-efficient residential buildings. We accomplished this using household-level data collected from 1128 households in 2016 in the HSCW zone in China. The average actual electricity consumption was found to be less than 1/5–1/3 of the regulation limit values. The heat pump air conditioners' actual schedules and setpoint settings, the energy efficiency labels of ACs, and the ventilation rates were found to be significant factors explaining the differences between theoretical and actual energy consumption.

Beyond identifying the EPG, our research also identified low and high consumers of electricity, which can help us to better understand the electricity consumption spectrum. The average annual household electricity consumption in 65%-BEES buildings was 2249 kWh/household. The yearly household electricity consumption in 50%-BEES buildings was 2132 kWh/household. In IBs, the average annual household electricity usage was 3018 kWh/household. The electricity Gini coefficient was 0.284. The lowest electricity-consuming households had common features, including are lower incomes, smaller floor areas, single or couple residents, and a lack of central air conditioning and electrical underfloor heating. Similarly, the common characteristics for the highest-consuming groups were higher incomes, larger floor areas, and extended families living in the households.

Our findings have important policy implications. First, the prebound effect exists in energy-efficient residential buildings in the hot summer and in the cold regions of China.

We should be concerned that a significant fraction of our national climate strategy rests on the types of policies that have not delivered on past promises. Second, our results can help policy-makers to understand that the benchmark for the assumption of building energy consumption values in building energy efficiency standards is a huge deviation from the actual situation. Therefore, the design values of building energy consumption regulated by the energy efficiency standards should be lower than the current values. The other important suggestion given to policy-makers is the energy consumption value regulation outlined in the 'Standard for energy consumption of building' (GB/T 51161-2016). Currently, the limitation value of electricity consumption for residential buildings in the HSCW zone is 3100 kWh/household. However, according to our survey, the maximum average electricity consumption was 2573 kWh/household. Therefore, in the future, when policy-makers revise this standard, the results of this research can offer them a foundation of benchmarking data for reference.

Although this study fills the gap in the literature by using actual household-level data to quantify the EPG between theoretical and actual consumption of energy-efficient residential buildings in China's HSCW climate zone, some limitations still exist. Due to individuals' unique energy use behavior and the sample size limit, the actual average energy consumption analyzed in this paper may not represent the actual energy consumption level in the HSCW climate zone in China. In future research, we will collect more detailed RBEUD to develop an energy consumption baseline for energy-efficient residential buildings. In addition, continuous data tracking can provide more accurate input parameters (closer to real-life energy use) for the purpose of building energy models, which will improve the accuracy of building energy prediction and narrow the EPG of energy-efficient residential buildings.

**Author Contributions:** Conceptualization, X.W.; methodology, X.W.; software, J.Y.; validation, K.Y.; formal analysis, X.W.; investigation, X.W., K.Y. and Z.L.; resources, X.W., K.Y. and Z.L.; data curation, J.Y.; writing—original draft preparation, X.W.; writing—review and editing, X.W.; visualization, Z.L.; supervision, X.M.; project administration, K.Y.; funding acquisition, X.M. All authors have read and agreed to the published version of the manuscript.

**Funding:** This research was funded by the National Key R&D Program of China, grant number 2018YFD1100203. And the Social Science Planning Project of Chongqing, grant number 2019QNGL30. And fundamental Research funds for the central universities, grant number SWU1909752.

**Conflicts of Interest:** The authors declare no conflict of interest.

## Abbreviations

| | |
|---|---|
| HSCW | Hot Summer and Cold Winter |
| IEA | International Energy Agency |
| RBEUD | Real building energy use data |
| EPG | Energy performance gap |
| ACs | Air Conditioners |
| BEES | Building Energy Efficiency Standards |
| 50%-BEES | Design Standard for Energy Efficiency 50% of Residential Buildings |
| 65%-BEES | Design Standard for Energy Efficiency 65% of Residential Buildings |
| CMCURD | Chongqing Municipal Commission of Urban-Rural Development |
| MOHURD | Ministry of Housing and Urban–Rural Development of the People's Republic of China |
| SGCEPC | State Grid Chongqing Electric Power Company |

**Appendix A**

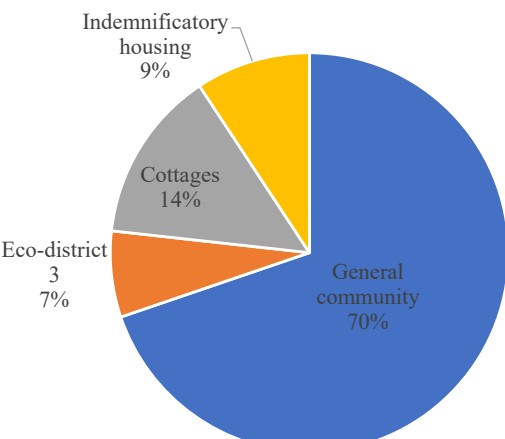

**Figure A1.** A Distribution of categories of the community of the survey samples (n = 1128).

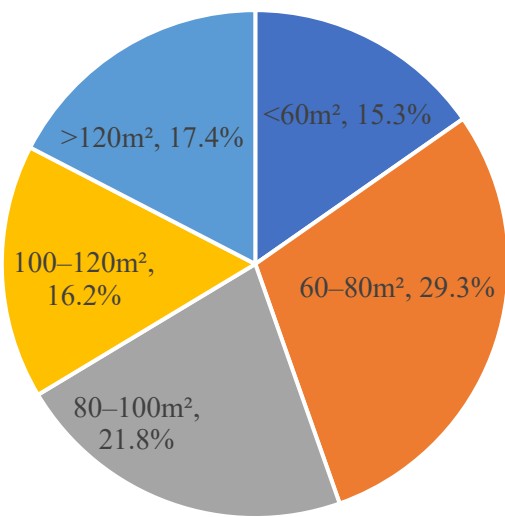

**Figure A2.** Distribution of household floor area in the survey data (n = 1128).

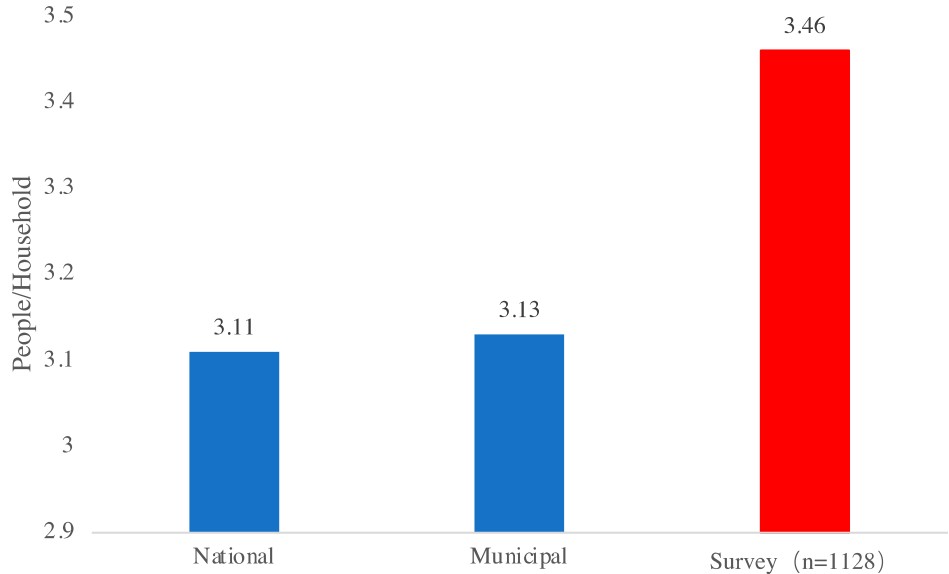

**Figure A3.** Average urban household size in China, Chongqing, and survey samples in 2016.

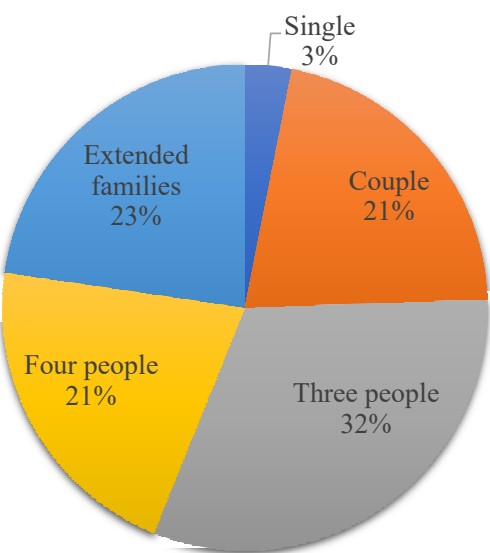

**Figure A4.** Family structure of urban households in Chongqing (n = 1128).

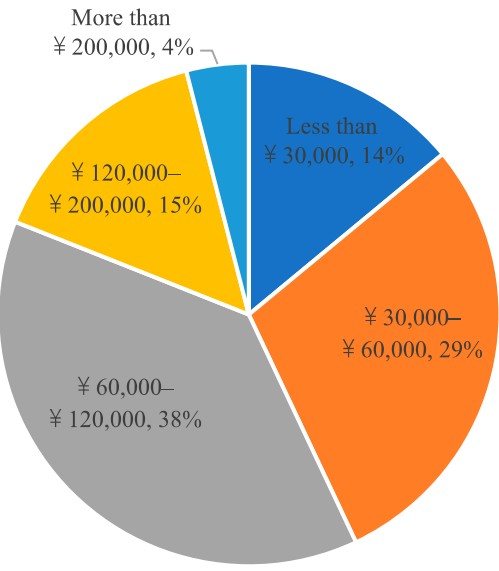

**Figure A5.** Percentage of annual income in the survey households (samples:1128).

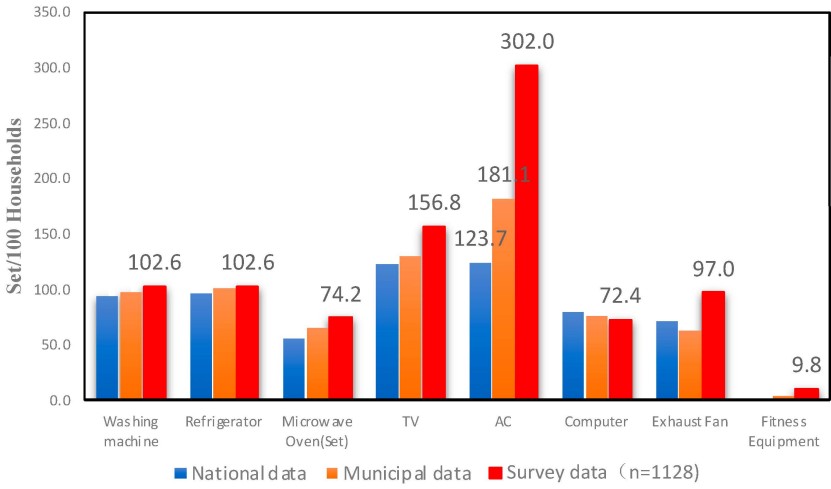

**Figure A6.** Ownership of major household appliances in China, Chongqing, and the survey population.

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
