# Peer review of "Using Real Building Energy Use Data to Explain the Energy Performance Gap of Energy-Efficient Residential Buildings: A Case Study from the Hot Summer and Cold Winter Zone in China"

_sustainability, doi:10.3390/su15021575_

Round 1

Reviewer 1 Report

This is an interesting topic about energy performance gap of energy-efficient residential buildings. The authors revealed that the actual 22 consumption of residential buildings is less than 1/5-1/3 of the theoretical limits where heat pump air explain most about the energy performance gap. A database of 1,128 households is also provided to determine the actual usage behavior parameters for building energy forecasting. However, here are some questions that I came up with:

1. This paper has two objectives: 1) gathering and analyzing ABEUD at the household level to identify and quantify the EPG in the residential sector, 2) understanding the role of occupants' behavior in explaining the EPG of energy-efficient residential buildings. However, in the introduction part, only the first objective is well illustrated.

2. The literature parts should focus more on occupant’s behavior.

3. What is the definition of real building energy use data? What exactly does it include? Is it measured only by energy consumption billing?

4. In line 166, “This database contains 508 residential buildings built with 65% level of building energy efficiency standards(65%-BEES) and 338 residential buildings built with 50% level 165 of building energy efficiency standards (50%-BEES).” The meaning of 65%-BEES and 50%-BEES is well explained.

5. In line 150, the details of questionnaire should be provided in the appendix. Or the authors could describe more about the questionnaire in “survey description” part.

6. In section 4, “residential buildings’ energy consumption is analyzed based on the actual consumption data with a focus on space heating and cooling”, how to obtain actual consumption of space heating and cooling. Is it obtained from the questionnaire or the smart meter used for space heating and cooling.

7. What is the definition of real building energy use data? What exactly does it include? Does it refer to the actual energy consumption value? If it also includes building characteristics, use behavior, etc., a definition is needed to avoid misunderstanding.

8. Proofreading is required and it would be better to correct the gramma errors throughout the manuscript, for example, in line 38, “it” should be capitalized.

Reviewer 2 Report

This study analyzes the causes of EPG from the perspective of actual household electricity consumption, considering household income, electricity consumption characteristics and other aspects. The article is well written, relevant data of residents in Chongqing are presented well and the gap between benchmarking for building energy consumption values assumption and the actual situation is discussed well. The research results have a good reference significance for future policy-making. I have a question: Is there any obvious difference between high-income households and low-income households in the average number of household member? Because the electricity consumption of high-income households is different from that of low-income households, if the number of household member is obviously different, the final data may need to be corrected.

Reviewer 3 Report

In line 199 and line 263, please change 238 kwh/household to 238 kWh/household.

In fig.9, the percentage for the lowest 10% electricity consumer in the first representation is not understood.
